Different revegetation types alter soil physical-chemical characteristics and fungal community in the Baishilazi Nature Reserve

Deng Jiaojiao 1 2
Yin You 1 3
Luo Jiyao 4
http://orcid.org/0000-0002-7367-3754 Zhu Wenxu 1 3 zhuwx@syau.edu.cn
Zhou Yongbin 1 3 yyzyb@163.com
1 College of Forestry, Shenyang Agriculture University , Shenyang , China
2 College of Land and Environment, Shenyang Agricultural University , Shenyang , China
3 Research Station of Liaohe-River Plain Forest Ecosystem, Chinese Forest Ecosystem Research Network, Shenyang Agricultural University , Tieling , China
4 Liaoning Baishi Lazi National Nature Reserve Administration , Dandong , China
Minasny Budiman
Electronic publication date: 2019 Jan 11
Publication date: 2019
Volume: 6
Electronic Location ID: e6251
Received 2018 Aug 30; Accepted 2018 Dec 8
Copyright: © 2018 Deng et al.
Copyright year: 2018
Copyright holder: Deng et al.
License: This is an open access article distributed under the terms of the Creative Commons Attribution License, which permits unrestricted use, distribution, reproduction and adaptation in any medium and for any purpose provided that it is properly attributed. For attribution, the original author(s), title, publication source (PeerJ) and either DOI or URL of the article must be cited.
License URL: https://creativecommons.org/licenses/by/4.0/

Keywords: Coniferous forest, Broadleaf forest, Fungal community diversity and composition, The Baishilazi Nature Reserve

Funding: National Key Research and Develepment Program 2017YFC050410501 Cultivation Plan for Youth Agricultural Science Technology Innovative Talents of Liaoning Province 2015047 Startup Foundation for Doctor of Liaoning 20170520064 Forest Scientific Research in the Public Welfare 201304216 This research was financially supported by the Sub-project of the National Key Research and Develepment Program (2017YFC050410501), the Cultivation Plan for Youth Agricultural Science and Technology Innovative Talents of Liaoning Province (2015047), the Startup Foundation for Doctor of Liaoning (20170520064) and the Special Fund for Forest Scientific Research in the Public Welfare (201304216). The funders had no role in study design, data collection and analysis, decision to publish, or preparation of the manuscript.

==============================
The effects of different revegetation types on soil physical–chemical characteristics and fungal community diversity and composition of soils sampled from five different revegetation types (JM, Juglans mandshurica; QM, Quercus mongolica; conifer-broadleaf forest (CB); LG, Larix gmelinii; PK, Pinus koraiensis) in the Baishilazi Nature Reserve were determined. Soil fungal communities were assessed employing ITS rRNA Illunima Miseq high-throughput sequencing. Responses of the soil fungi community to soil environmental factors were assessed through canonical correspondence analysis (CCA) and Pearson correlation analysis. The coniferous forests (L. gmelinii, P. koraiensis) and CB had reduced soil total carbon (C), total nitrogen (N), and available nitrogen (AN) values compared with the broadleaf forest (J. mandshurica, Q. mongolica). The average fungus diversity according to the Shannon, ACE, Chao1, and Simpson index were increased in the J. mandshurica site. Basidiomycota, Ascomycota, Zygomycota, and Rozellomycota were the dominant fungal taxa in this region. The phylum Basidiomycota was dominant in the Q. mongolica, CB, L. gmelinii, and P. koraiensis sites, while Ascomycota was the dominant phylum in the J. mandshurica site. The clear differentiation of fungal communities and the clustering in the heatmap and in non-metric multidimensional scaling plot showed that broadleaf forests, CB, and coniferous forests harbored different fungal communities. The results of the CCA showed that soil environmental factors, such as soil pH, total C, total N, AN, and available phosphorus (P) greatly influenced the fungal community structure. Based on our results, the different responses of the soil fungal communities to the different revegetation types largely dependent on different forest types and soil physicochemical characteristic in Baishilazi Nature Reserve.

Introduction

Due to long-term human disturbances and intensive land use, the native vegetation of temperate zones in China is severely damaged, with reduced biodiversity and deteriorated ecological functions (Liu & Diamond, 2005). Numerous researches have established that vegetation restoration is an important measure to obtain ecological benefits (Gao et al., 2002; Nunezmir et al., 2015; Xiao et al., 2015), such as enhancing biodiversity (Dosskey, Bentrup & Schoeneberger, 2012), the restoration of damaged natural ecological system (Fulé et al., 2012), and the recovery of ecosystem services (Benayas et al., 2009). Revegetation also has numerous positive effects on soil physicochemical characteristics, such as soil bulk density, field capacity (Zhang et al., 2018), infiltration rates (Wu et al., 2016), soil organic carbon (Georgiadis et al., 2017), and soil nitrogen (Jin et al., 2016). Feedback processes of plant-soil play crucial roles in altering the structure and dynamics of soil microorganisms (Herrera Paredes & Lebeis, 2016). The soil microbial community, which is a key bridge that connects the plant community with soil processes, is one of the most important regulators of soil nutrient transformation (Cheng et al., 2013). Soil microorganisms can not only directly affect the storage of soil nutrients via microbial biomass, but also can indirectly effect soil nutrient transformation through the metabolic activity (Jangid et al., 2013; You et al., 2014). In this sense, different revegetation types, combined with accurate biological monitoring, can achieve effective, and targeted restoration goals (Collen & Nicholson, 2014). However, studies on the changes in soil microbial community dynamics, despite the important role of microorganisms in biogeochemical cycling, are still scarce (Guo et al., 2018).

Fungal community and diversity have important influences on plant communities and ecosystems (Van Der Heijden, Bardgett & Van Straalen, 2008; Devi et al., 2012). Furthermore, fungi play crucial roles in many respects of ecosystem development (Chen et al., 2010; Geml et al., 2014), determining biochemical cycles in continental ecosystems (Tedersoo et al., 2014). Fungal diversity and community composition are closely related to numerous abiotic and biotic factors, such as elevation (Kernaghan & Harper, 2001; Bahram et al., 2012), soil environment (Peter, Ayer & Egli, 2001; Dickie, Xu & Koide, 2002), plant species (Lovett et al., 2004; Weand et al., 2010), plant diversity (Dickie, 2007; Waldrop et al., 2006), and stand age (Zhu et al., 2010; Wallander et al., 2010). An increasing number of studies have shown that a number of soil properties, including soil pH (Fierer et al., 2005), soil texture (Girvan et al., 2003), and soil nitrogen availability (Frey et al., 2004), can be associated with changes in fungal community structure; these soil characteristics are often influenced by vegetation cover at the same area. Any changes in the fungal community in the process of ecological restoration are key indicators of restoration success (Harris, 2009). A previous study has reported that replanting native vegetation can result in dramatic shifts in the fungal community toward that of the natural fungal community (Yan et al., 2018). Unfortunately, few researchers have addressed the connection between the different revegetation types and the fungal community structure in broadleaf forests, coniferous forests, and conifer-broadleaf forest (CB). Specifically, it is unclear which soil environmental factors play the principal roles in driving fungal dynamics. Previous work has concentrated on the relationships between soil characteristics and soil fungi under grassland and leguminous species (Harrison & Bardgett, 2010), but there remains a need for interpreting which of these factors were the dominant influence on the soil fungal communities in different revegetation types.

As the national nature reserves, the Baishilazi Nature Reserve is located in the mountainous region of the eastern Liaoning Province, China. The reserve was established in 1988 and is part of the Changbai Mountain system. The original vegetation was broadleaf Pinus koraiensis forests, which were severely damaged due to the over-exploitation of the past 100 years. At present, the vegetation mainly consists of natural secondary forests and coniferous forests, which provides a unique opportunity to investigate the soil fungal communities of different revegetation ecosystems under the same climatic conditions. Numerous researches have investigated the changes in soil microbial biomass (Fan & Liu, 2014), and soil organic carbon contents (Qi, 2017) in different revegetation types; however, studies on the impacts of different reforestation pathways on the soil fungal community are scarce. In this context, we applied pyrosequencing of the ITS rRNA gene to explore both the diversity and composition of soil fungal communities in responses to different revegetation types from five sites in the Baishilazi Nature Reserve, Liaoning Province, China. Our objective was to examine how soil fungi may respond to different revegetation types and, more specifically, how the abundance and composition of soil fungal communities respond to changes in soil physicochemical properties.

Material and Methods

Site description

The field study was carried out at the Baishilazi Nature Reserve (approval number# 20170628-7), the eastern mountainous areas of Liaoning Province (40.83°–40.95°N, 124.74°–124.96°E; Fig. 1). It is a comprehensive nature reserve with forest ecosystem as the main protection object. The reserve covers an area of 7,407 ha and belongs to the Changbai mountain range. This area is characterized by a continental monsoon climate, with long cold winters, warm wet summers, and a higher diurnal temperature variation. The annual mean amount of evaporation is 885 mm, with an annual average temperature of 6.4 °C, and an average annual precipitation of 1,158 mm. The soil type is Brown soil (Zhang et al., 2017a). The region has a relatively rich and unique biodiversity, possessing significant ecological status and scientific value both in China and on a global level. The characteristics of the five selected study samples are listed in Table 1.

Figure 1 Map showing sampling location of the study.

(A) Study site; (B) Larix gmelinii; (C) Pinus Koraiensis; (D) Conifer-broadleaf forest; (E) Juglans mandshurica; (F) Quercus mongolica.

Table 1 Sites information of the Baishilazi Nature Reserve.

Vegetation types	Dominant species	Elevation (m above sea level)	Forest type	
JM	Acanthopanax senticosus, Padus racemose, Magnolia sieboldii, Pimpinella brachycarpa, Puccinellia tenuiflora	901.8	Natural secondary forest	
QM	Acer mono, Cerasus tomentosa, Carpinus cordata	842.3	Natural secondary forest	
CB	Betula ermanii, Pinus koraiensis, Schisandra chinensis, Phryma leptostachya L. subsp. asiatica	826.5	Natural secondary forest	
LG	Daemonorops margaritae	552.7	Plantation forest	
PK	Daemonorops margaritae, Pteridium aquilinum	552.7	Plantation forest	
Note:

JM, Juglans mandshurica; QM, Quercus mongolica; CB, Conifer-broadleaf forest; LG, Larix gmelinii; PK, Pinus koraiensis.

Soil sampling

In July 2017, we sampled soils from three randomly selected 20 × 20 m plots per forest type after removal of the litter layer. The sampled forest types were including Juglans mandshurica (JM), Quercus mongolica (QM), CB, Larix gmelinii (LG), and Pinus koraiensis (PK), giving a total of 15 plots. Soil samples were collected use of a soil auger with eight cm in diameter, and 10 cm deep. The soils of 15–20 points were collected at a depth of 0–10 cm along an “S” shaped path to ensure the representativeness of soil samples in each forest, mixed together and placed in sterilized ziplock bags as a replicate sample. Identically, in each revegetation form, three subsamples were collected. Immediately arrival to the laboratory the samples were stored cooled boxes, sieved (two mm mesh) to undock roots and dopant, and divided into two sub-samples, of which one was air-dried and used for physical and chemical analyses, and the other one was stored at −80 °C prior to DNA extraction and used for microbial analyses.

Determination of the physical and chemical properties

Soil pH was measured using a pH meter after shaking a soil-water (1:5 w/v) suspension for 30 min (Bao, 2000; Ren et al., 2016). Soil total C and total N were measured with an Elemental Analyzer (Elementar, Langenselbold, Germany) (Schrumpf et al., 2011). Available nitrogen (AN) was determined by the alkali diffusion method (Bao, 2000). Total phosphorus was measured by spectrophotometry after wet digestion with HClO4–H2SO4 (Parkinson & Allen, 2009). Available phosphorus (AP) was measured using the colorimetric method with extraction via 0.5M NaHCO3 (Emteryd, 1989).

DNA extraction and PCR amplification sequencing

Total fungal genomic DNA samples were extracted from 0.5 g of soil using the Fast DNA SPIN extraction kits (MP Biomedicals, Santa Ana, CA, USA), according to the manufacturer’s instructions. The quantity and quality of extracted DNAs were measured using a NanoDrop ND-1000 spectrophotometer (Thermo Fisher Scientific, Waltham, MA, USA). The primer sets: ITS1F (5′-CTTGGTCATTTAGAGGAAGTAA-3′) (Gardes & Bruns, 1993) and ITS2F (5′-GCTGCGTTCTTCATCGATGC-3′) (White et al., 1990) were selected to target the fungal ITS1 region. Sample-specific seven-bp barcodes were incorporated into the primers for multiplex sequencing. The PCR amplification required two steps. During the first step, each of three independent 25 μl reactions per DNA sample included five μl of Q5 reaction buffer (5×), five μl of Q5 High-Fidelity GC buffer (5×), one μl (10 uM) of each forward and reverse primer, two μl (2.5 mM) of dNTPs, 0.25 μl of Q5 High-Fidelity DNA Polymerase (five U/μl), two μl of DNA Template, and 8.75 μl of ddH2O. Cycling conditions were 98 °C for 5 min; 25 cycles of 98 °C for 15 s, 55 °C for 30 s, 72 °C for 30 s, followed by 72 °C for 5 min. The PCR amplicons were purified with Agencourt AMPure Beads (Beckman Coulter, Indianapolis, IN, USA) and quantified using the PicoGreen dsDNA Assay Kit (Invitrogen, Carlsbad, CA, USA). After the individual quantification step, amplicons were pooled at equal amounts, and pair-end 2 × 300 bp sequencing was performed using the Illlumina MiSeq platform with the MiSeq Reagent Kit v3.

Sequence data analysis

The raw data yielded from Illumina sequencing were analyzed using QIIME software (v1.9.0) and the UPARSE pipeline (Zhong, Yan & Shangguan, 2015). The UPARSE pipeline was performed for taxonomic assignment with similarities >97% (Edgar, 2013). Taxonomic classification was conducted with Unite databases for fungi. The raw data of fungi were submitted to the NCBI with the SRA accession number: PRJNA494279.

The operational taxonomic identity was annotated using a BLAST algorithm against sequences within the Unite Database, using the QIIME software (Kõljalg et al., 2013). An Operational taxonomic units (OTUs) table was further generated to record the abundance of each OTU in each sample and the taxonomy of these OTUs. OTU-level alpha diversity indices, such as Chao1 index (Chao & Bunge, 2002), ACE index, Shannon index, and Simpson index, were computed using the OTU table in QIIME (Caporaso et al., 2010).

Statistical analysis

The shared and unique OTUs among samples were used to generate Venn diagrams using the R v3.3.2 (R Development Core Team, 2017) package with “VennDiagram,” based on the occurrence of OTUs across sample (Zaura et al., 2009). The heatmap representation of the top 50 classified genera in per sample was built using the R v3.2.2 package with “gplot” and “pheatmap” (R Development Core Team, 2017). Non-metric multidimensional scaling (NMDS) with the Bray–Curtis matrices was conducted the soil fungal community composition using the “metaMDS” function in the Vegan package (Yang et al., 2017). The linear discriminant analysis (LDA) effect size (LEfSe) method were performed to detect potential biomarkers of abundant taxa based on a normalized relative abundance matrix across groups, using the default parameters, which was built using Galaxy based on the online interactive analysis of microflora data (Segata et al., 2011).

Differences in soil physicochemical characteristics, fungal alpha diversity indices, and the relative abundance of taxa (phyla and genus) of different forest soils were compared using one-way analysis of variance, which was followed by the least significant difference test performed in IBM SPSS (version 19.0; Chicago, IL, USA) (Banerjee et al., 2016). In order to find the tree species affecting the soil quality, principal component analysis were used to analyze all parameters via Canoco 4.5 (Braak & Smilauer, 2002). Pearson correlation analysis was used to evaluate the correlations between soil fungal community diversity, fungal community composition, and soil characteristics. Canonical correspondence analysis, which was performed via Canoco 4.5, was used to evaluate the linkages between dominant fungal groups and soil measured environmental factors (Braak & Smilauer, 2002).

Results

Soil physicochemical properties

As seen in Table 2, soil pH value ranged from 4.89 to 5.70. Soil pH value under Q. mongolica was the most acidic with 4.89, followed by CB, while, J. mandshurica contained the highest soil pH value. There were significant differences among different forest types regarding soil total C and total N contents (F = 112.275, P < 0.001; F = 29.72, P < 0.001; Table 2). Interestingly, both total C and total N exhibited highest value in the soil of J. mandshurica, with values of 100.53 and 7.80 g/kg, but only 41.70 and 3.58 g/kg in the P. koraiensis site, respectively. Soil C/N values in all the treatments were below 25:1, among which CB had the highest C/N with 13.81. AN was found in ranked order of J. mandshurica > Q. mongolica > CB >L. gmelinii > P. koraiensis. There were also significant differences in total P among different forest types (F = 26.59, P < 0.001; Table 2), with the highest values under CB (Table 2).

Table 2 Soil physical and chemical properties of different revegetation types.

Chemical features	JM	QM	CB	LG	PK	F test	P-value	
pH	5.70a	4.89c	4.99c	5.40b	5.48b	111.80	P < 0.001	
Total C (g/kg)	100.53a	84.62b	75.49c	43.79d	41.70d	112.28	P < 0.001	
Total N (g/kg)	7.80a	7.38a	5.47b	3.85c	3.58c	29.72	P < 0.001	
C/N	12.89ab	11.64bc	13.81a	11.36c	11.65bc	6.50	P = 0.008	
Total P (g/kg)	0.93b	0.74bc	1.46a	0.62c	0.77bc	26.59	P < 0.001	
Available N (mg/kg)	57.15a	41.25b	43.60b	33.35c	28.04c	14.99	P < 0.001	
Available P (mg/kg)	4.42ab	2.39ab	2.53ab	1.21b	5.65a	1.99	P = 0.172	
Note:

Results from the ANOVAs are included (F test and P-value). Different letters in the same line (a, b, c) indicate a significant difference at P < 0.05. See Table 1 for abbreviations.

The first two axes of the principal components analysis accounted for 96.9% of the total variance. The biplot showed a clear spatial separation among different revegetation types. In fact, the axis 1 discriminated for the P. koraiensis situated in the first quadrant and P. koraiensis, Q. mongolica, and L. gmelinii in third and fourth quadrant, while the axis 2 discriminated for P. koraiensis and L. gmelinii situated in the first and second quadrants and the J. mandshurica, Q. mongolica, and CM soils in the second and third quadrants. Also, the investigated soil characteristics were clearly separated in the quadrants (Fig. 2). The C/N was situated in the first quadrant, soil total C, and AP concentrations in the second quadrant, soil total P and total N in the third quadrant, and all other parameters were located in the fourth quadrant (Fig. 2).

Figure 2 Results of principle component analysis-Biplot of the all investigated paramenters.

JM, Juglans mandshurica; QM, Quercus mongolica; CB, Conifer-broadleaf forest; LG, Larix gmelinii; PK, Pinus koraiensis. TN, total N; TC, total C; TP, total P; AN, available N; AP, available P.

Fungal community diversity responses to different revegetation types

Fungal α-diversity varied greatly across the samples. The Shannon index, ACE index, Chao1 index, and Simpson index showed the highest values in the J. mandshurica site, with 8.18, 879.57, 879.08, and 0.99, respectively, followed by the CB site. The ACE index and Chao1 index were the lowest in the L. gmelinii site, with 522.15 and 521.58, respectively, while, the Shannon index and Simpson index were the lowest in the Q. mongolica site, with 5.79 and 0.89, respectively (Table 3). Pearson correlation analysis indicated that the Simpson index (r = 0.680, P < 0.01) and Shannon index (r = 0.659, P < 0.01) were positively correlated with soil pH, while the Shannon index was positively correlated with C/N (r = 0.528, P < 0.05). In addition, ACE index and Chao1 index were significantly positively correlated with total C (P < 0.05), C/N and total P (P < 0.01) (Table 4).

Table 3 Soil fungal diversity indexes of different revegetation types.

Types	Number of sequences	OTUs number (phylum)	Shannon index	ACE index	Chao1 index	Simpson index	
JM	40,811	715	8.18 ± 0.23a	879.57 ± 64.4767a	879.08 ± 64.48a	0.99 ± 0.00a	
QM	33,752	518	5.79 ± 0.28c	597.78 ± 98.62b	598.00 ± 98.89b	0.89 ± 0.03b	
CB	37,669	743	7.18 ± 0.34b	870.95 ± 192.83a	866.17 ± 184.59a	0.98 ± 0.01a	
LG	37,959	455	6.49 ± 0.22bc	522.15 ± 100.95b	521.58 ± 101.48b	0.97 ± 0.01a	
PK	63,447	525	7.06 ± 0.89b	650.67 ± 108.58b	649.10 ± 109.27b	0.97 ± 0.02a	
Notes:

Different letters in the same line (a, b, c) indicate a significant difference at P < 0.05.

JM, Juglans mandshurica; QM, Quercus mongolica; CB, Conifer-broadleaf forest; LG, Larix gmelinii; PK, Pinus koraiensis.

Table 4 Person’s rank correlation coefficients between fungi diversity indices and measured soil characteristics.

	pH	Total C	Total N	C/N	Available N	Total P	Available P	
Simpson	0.680**	−0.139	−0.337	0.472	−0.043	0.334	0.297	
Chao1	0.089	0.573*	0.389	0.715**	0.397	0.725**	0.238	
ACE	0.085	0.567*	0.383	0.714**	0.389	0.730**	0.234	
Shannon	0.659**	0.302	0.132	0.528*	0.312	0.361	0.508	
Notes:

* Correlation is significant at the 0.05 level (one-tailed).

** Correlation is significant at the 0.01 level (two-tailed).

Fungal community structure responses to different revegetation types

A total of 640,914 high-quality ITS sequences were obtained after the elimination of chimeras and sequences of low quality, with an average of 42,727 sequences being obtained in each soil sample. At the phylum level, we found 8,875 fungal OTUs after quality filtering. On average, 592 OTUs were found in each sample. A maximum of 743 OTUs were detected in the CB site, however, only 455 OTUs were obtained in the L. gmelinii site (Table 3). In order to determine rarefaction curves, richness, and diversity, 22,716 reads were randomly selected from each sample. At the 3% dissimilarity level (Fig. 3), the curve tended to flatten with the number of measured sequences increases, indicating that the experiment had obtained most of the sample information and had been able to reflect the fungal community composition of the forest soil.

Figure 3 Rarefaction curves.

JM, Juglans mandshurica; QM, Quercus mongolica; CB, Conifer-broadleaf forest; LG, Larix gmelinii; PK, Pinus koraiensis.

The obtained sequences were affiliated with 15 phyla (including unknown). The dominant phyla, accounting for more than 1% of the overall communities, were Basidiomycota, Ascomycota, Zygomycota, and Rozellomycota, with relative abundance values ranging from 21.31 to 66.08%, 24.82 to 51.88%, 2.21 to 6.37%, and 0.42 to 2.09%, respectively (Fig. 4). Phyla included Cercozoa, Chytridiomycota, Glomeromycota, and Ciliophora, which were less abundant (<1% of all classified sequences), but still were found in all of the examined soils. The relative abundances of these most abundant fungal phyla varied significantly among different forest types. The relative abundance of Ascomycota in J. mandshurica was significantly higher than that of the other phyla, while the relative abundances of Basidiomycota was the lowest in J. mandshurica. In the CB site, the relative abundance of Basidiomycota was highest, while, the relative abundances of Ascomycota and Rozellomycota were the lowest (Fig. 4).

Figure 4 Relative abundance of fungus phyla present in five different revegetation types.

JM, Juglans mandshurica; QM, Quercus mongolica; CB, Conifer-broadleaf forest; LG, Larix gmelinii; PK, Pinus koraiensis.

At the genus level, the dominant genus, accounting for more than 1% of the overall communities, were Sebacina (5.95%), Russula (4.38%), Tomentella (3.74%), Mortierella (2.97%), Trechispora (2.17%), Piloderma (1.92%), Humicola (1.75%), Suillus (1.69%), Geminibasidium (1.65%), Ramaria (1.62%), Archaeorhizomyces (1.59%), Cryptococcus (1.55%), Simplicillium (1.50%), Oidiodendron (1.37%), Inocybe (1.32%), Basidiobolus (1.20%), and Bullera (1.07%) (Fig. S1). Sebacina was the most abundant genus at P. koraiensis, accounting for 21.17%. The relative abundances of Russula showed highest in the CB than others (Fig. S1).

Venn diagrams were used to compare the fungal communities based on shared and unique OTUs among the samples. At the genus level, the Venn diagram showed 110 OTUs among the five forest types (Fig. 5). A total of 1,453, 1,006, 1,321, 1,143, and 1,742 OTUs were observed in the CB, L. gmelinii, P. koraiensis, Q. mongolica, and J. mandshurica (Fig. 5).

Figure 5 Venn diagrams of OUT richness.

JM, Juglans mandshurica; QM, Quercus mongolica; CB, Conifer-broadleaf forest; LG, Larix gmelinii; PK, Pinus koraiensis.

To illustrate the fungal community structures of J. mandshurica, Q. mongolica, CB, L. gmelinii, and P. koraiensis, the heatmap analysis based on the top 50 most abundant fungal genera was used to intuitively display the differences in relative abundances of fungal OTUs among the samples (Fig. 6). The relative abundance of the soil fungal community from high to low is represented by red through black to green, reflecting the different compositions and relative abundances of soil fungi under different forest types. The genera Tomentella, Piloderma, Suillus, Oidiodendron, Inocybe, Entoloma, Cortinarius, Helvellosebacina, and Phaeoacremonium dominated in L. gmelinii. While in CB, Russula, Trichoderma, Leucoagaricus, Amphinema, Umbelopsis, and Thelephora were the most dominated genera. Cladophialophora, Byssocorticium, Trichoderma, Hygrocybe, Exophiala, Leotia, and Knufia dominated in Q. mongolica. Correspondingly, NMDS based Bray distance was carried out to show the distinct separation among different forest types (Fig. 7). By considering the phylogenetic relationship, the heatmap and NMDS plot elucidated that distinct variations of fungal community structure occurred among the different revegetation types, demonstrating the different effects of tree species on the fungal community composition following different revegetation types.

Figure 6 Heatmap and hierarchical cluster analysis based on the relative abundances of the top 50 genera identified in the bacterial communities of the soils.

JM, Juglans mandshurica; QM, Quercus mongolica; CB, Conifer-broadleaf forest; LG, Larix gmelinii; PK, Pinus koraiensis.

Figure 7 Weighted UniFrac NMDS analysis of the composition of fungal communities in the soil of forests with different dominant trees.

JM, Juglans mandshurica; QM, Quercus mongolica; CB, Conifer-broadleaf forest; LG, Larix gmelinii; PK, Pinus koraiensis.

The LEfSe analysis was documented to determine the classified fungal taxa with significant abundance differences among the different sampling sites. As presented in Fig. 8, 16 fungal taxa were significantly different among the sites with LDA effect size scores were >4.8 (Fig. 8A), and five fungal taxa were showed significantly different with LDA effect size scores were >5.2 (Fig. 8B). At the phylum level, the biomarkers were affiliated with Basidiomycota, and Ascomycota, respectively.

Figure 8 The cladogram of fungal communities among different sampling sites with LDA effect size scores were >4.8 (A), and >5.2 (B). In the cladogram, the circles radiating represent fungal taxon from phylum to genus from the inside.

JM, Juglans mandshurica; QM, Quercus mongolica; CB, Conifer-broadleaf forest; LG, Larix gmelinii; PK, Pinus koraiensis.

Fungal community distribution as related to the soil properties

Canonical correspondence analysis was used to analyze the relative abundances of dominant fungal phyla constrained by soil properties (Fig. 9). The results showed that the cumulative interpretation variations of the first and second axes were 93.5%, indicating that soil environmental factors greatly influenced the fungal community structure. At the phylum level (Fig. 9), soil pH (r = 0.9104) and AP (r = 0.6891) were significantly correlated with axis 1, and the first axial interpretation rate was 69.1%. The parameters C/N (r = −0.7322) and total P (r = −0.8094) were significantly related with axis 2.

Figure 9 Canonical correspondence analysis (CCA) on soil dominant fungal phyla constrained by soil variables.

TN, total N; TC, total C; TP, total P; AN, available N; AP, available P.

Pearson correlation analysis were used to explore the relationships between soil properties and the relative abundances of the four most abundant fungal phyla and 15 most abundant fungal genera. At the phylum level, the relative abundance of Basidiomycota was significantly negatively correlated with soil pH (r = −0.680, P < 0.01) and AP (r = −0.611, P < 0.01). Ascomycota relative abundance was positively correlated with total C (r = 0.608, P < 0.05), total N (r = 0.655, P < 0.01) and AN (r = 0.693, P < 0.01). Zygomycota relative abundance was positively correlated with pH (r = 0.530, P < 0.05) and AP (r = 0.665, P < 0.01). Rozellomycota relative abundance was significantly positively correlated with pH (r = 0.716, P < 0.01; Table 5).

Table 5 Person’s rank correlations between the relative abundances of dominant bacteria groups and available edaphic factors.

Fungal group	pH	Total C	Total N	C/N	Available N	Total P	Available P	
Phylun	–	–	–	–	–	–	–	
Basidiomycota	−0.680**	−0.455	−0.466	−0.010	−0.506	0.107	−0.611**	
Ascomycota	0.416	0.608*	0.655**	−0.004	0.693**	−0.058	0.462	
Zygomycota	0.530*	0.274	0.179	0.284	0.245	0.146	0.665**	
Rozellomycota	0.716**	−0.016	0.036	−0.278	0.052	−0.460	0.002	
Genus	–	–	–	–	–	–	–	
Sebacina	0.192	−0.563*	−0.533*	−0.282	−0.604*	−0.132	0.171	
Russula	−0.491	0.170	0.025	0.637*	0.171	0.751**	−0.121	
Tomentella	0.009	−0.407	−0.339	−0.371	−0.247	−0.360	−0.473	
Mortierella	0.524*	0.456	0.383	0.264	0.447	0.123	0.623*	
Trechispora	−0.182	0.096	−0.039	0.490	0.071	0.593*	−0.415	
Piloderma	−0.066	−0.378	−0.278	−0.452	−0.316	−0.534*	−0.482	
Humicola	0.302	0.745**	0.735**	0.146	0.705**	−0.098	0.152	
Suillus	0.118	−0.438	−0.402	−0.287	−0.263	−0.236	−0.266	
Geminibasidium	0.252	−0.530*	−0.516*	−0.226	−0.569*	−0.105	0.271	
Ramaria	0.116	−0.420	−0.378	−0.303	−0.353	−0.402	−0.368	
Archaeorhizomyces	−0.111	0.300	0.132	0.672**	0.334	0.591*	−0.057	
Cryptococcus	0.114	−0.655**	−0.655**	−0.212	−0.698**	−0.069	−0.031	
Simplicillium	0.540*	0.415	0.380	0.110	0.387	−0.009	0.509	
Oidiodendron	0.024	−0.417	−0.398	−0.211	−0.261	−0.098	−0.338	
Inocybe	0.144	−0.518*	−0.459	−0.397	−0.340	−0.383	−0.380	
Basidiobolus	0.431	−0.378	−0.434	0.023	−0.406	−0.143	0.578*	
Bullera	0.596*	0.340	0.273	0.231	0.313	0.091	0.357	
Notes:

* Correlation significant at 0.05 level (two-tailed).

** Correlation significant at 0.01 level (two-tailed).

At the genus level, the abundance of Sebacina was significantly negatively correlated with total C (r = −0.563, P < 0.05), total N (r = −0.533, P < 0.05) and AN (r = −0.604, P < 0.05). Russula was significantly positively correlated with C/N (r = 0.637, P < 0.05) and total P (r = 0.751, P < 0.01). While, Humicola was significantly positively correlated with total C (r = 0.745, P < 0.01), total N (r = 0.735, P < 0.01), and AN (r = 0.705, P < 0.01). The relative abundance of Geminibasidium was significantly negatively correlated with total C (r = −0.530, P < 0.05), total N (r = −0.516, P < 0.05), and AN (r = −0.569, P < 0.05). Archaeorhizomyces exhibited a positive correlation with C/N (r = 0.672, P < 0.01) and total P (r = 0.591, P < 0.05). Cryptococcus abundance showed a significantly negative correlation with total C (r = −0.655, P < 0.01), total N (r = −0.655, P < 0.01), and AN (r = −0.698, P < 0.01). While, Simplicillium (r = 0.540, P < 0.05), and Bullera (r = 0.596, P < 0.05) were significantly negatively correlated with pH.

Discussion

Soil characteristics of the different revegetation types

The soil nutrient concentrations (C, N, and P) we observed varied significantly among the different revegetation types (Table 2). According to our findings, coniferous forests (L. gmelinii, P. koraiensis) and the CB had reduced soil total C, total N, and AN value compared with broadleaf forests (J. mandshurica, Q. mongolica), which was consistent with the study of Rahimabady, Akbarinia & Kooch (2015). This influence may be attributed to different tree species with different litter quality and root exudates (Grayston & Prescott, 2005). In our study, soil total C, total N, and AN in L. gmelinii were higher than those of P. koraiensis. A previous study has indicated that L. gmelinii is a cold temperate deciduous coniferous forest with a higher litter amount. Despite the numerous recalcitrant substances found in the litter from L. gmelinii site, such as lignin, resin, tannin, and wax, the dense coniferous litter covering the soil surface impedes air circulation, and accelerates the accumulation of soil nutrients. On the contrary, P. koraiensis is often a component of warm evergreen coniferous forests; it produces relatively little litter, and its nutrient content in the soil is relatively low (Yang & Han, 2001). Soil pH in this area ranged from 4.89 to 5.70, and compared to others, the soil under Q. mongolica was most acidity, which might be associated with the litter quality. Compared to other broadleaf forests, the Q. mongolica litter leaf quality is low, which has low nitrogen content, high C/N ratio, higher lignin content, and higher lignin/N (Gao, Kang & Han, 2016). Therefore, these are several factors contributing to the effect that soil pH under Q. mongolica was lowest. In fact, it can be said that different tree species result in different soil characteristics under the same climatic conditions because of differences in litter quality and quantity (Jahed, Hosseini & Kooch, 2014), which was also observed in our study that the separation of the soils into four groups could depend on the fact that each one is mainly affected by certain soil characteristics (Fig. 2). The forest types would seem to play an important role in regulating soil characteristics inside the same soil climate and, in particular, the difference between broadleaf forest and coniferous forest should be noted.

Fungal community diversity and structure response to different revegetation types

We documented that different forest revegetation types had distinct soil fungal community diversity and composition (Table 3; Fig. 4; Fig. S1), as reported by Myers et al. (2001). In our study, we observed that the average fungal Shannon index, ACE index, Chao1 index, and Simpson index were the highest in J. mandshurica, followed by CB (Table 3), indicating that soil fungal richness and evenness indices in J. mandshurica were the highest. The ACE index and Chao1 index were the lowest in the L. gmelinii while, the Shannon index and Simpson index were the lowest in the Q. mongolica. This finding may be attributed to the differences in the chemical composition and decomposition rate of litter (Bray, Kitajima & Mack, 2012), which modify soil physical and chemical properties and, consequently, alter the soil fungal diversity indices. These findings verified that the soil fungal diversity indices were affected by tree species following revegetation.

In terms of phyla and genera, the composition of fungal community did not significantly differ among the different revegetation types; however, the relative abundance values varied, probably as a result of different root residues and secretions produced by different tree species (Degrune et al., 2015). Although our research is limited due to the low number of replicates (three for each kind of revegetation forest), significant differences in fungal community relative abundance among different revegetation types were observed. The results of our comparison of soil fungal communities among different revegetation types revealed that the predominant taxa of fungal communities were the phylum Basidiomycota in the Q. mongolica, CB, L. gmelinii, and P. koraiensis sites, followed by Ascomycota, Zygomycota, and Rozellomycota (Fig. 4), which was consistent with the results from Gutianshan National Nature Reserve (Yu et al., 2013), and from Mount Nadu, southwestern China (Liu et al., 2018b). Similar studies have also found such results (Leff et al., 2015; Yu et al., 2013). Basidiomycota tended to live in dry and cooler environments (Treseder et al., 2014), and the relative abundance of Basidiomycete in soils might be related to their ability to degrade of lignocellulose (Lundell, Makela & Hildén, 2010), which were affected by dynamics of soil organic matter (Hannula et al., 2012). On the contrary, we observed that the relative abundance of Ascomycota, over Basidiomycota, Zygomycota, and Rozellomycota, was predominant group in the J. mandshurica, which was in agreement with previous research (Curlevski et al., 2010). Findings from other tropical regions indicated that in the broadleaf forests, Ascomycota was the most predominant phylum (Kerfahi et al., 2014; Ritter et al., 2018). In our study, the higher abundance of Ascomycota in J. mandshurica suggested the enrichment of saprotrophic species, proving that Ascomycota tend to use the easily degradable residues (Lundell, Makela & Hildén, 2010), which might be related to organic matter input (Lundell, Makela & Hildén, 2010). However, the finding from Yarraman showed that Zygomycota was the dominant phylum (He, Xu & Hughes, 2005). These disparate results may indicate that both environmental filtering and niche differentiation determine the global distributions of soil fungi (Bahram et al., 2018).

The dominant fungal genera (Sebacina, Russula, Tomentella) were representative of the dominant genera found in our study (Fig. S1), which was accordant with previous research (Welc et al., 2014). Sebacina was the most common genus in our study, and previous studies have put forward that Sebacina could help its host plant to overcome biotic and abiotic stresses by supplying it with water and nutrients (Gao & Yang, 2016). Based on previous research, Tomentella has been reported to be distributed throughout the world (Kõljalg et al., 2000), which was also the common genus in our study. The influence of different revegetation types on the soil fungal community is often related to the nature and quantity of organic matter returned by plant litter, which provides major resources for soil microorganisms (Saetre & Bååth, 2000; Wardle et al., 2004).

The results of clear differentiation in the heatmap (Fig. 6) and NMDS (Fig. 7) plots illustrated that distinct differences in the fungal communities were observed in different revegetation types, suggesting that broad-leaved forests and coniferous forests each owned different fungal community. Our results were agreement with previous study which have established that the composition of the soil fungal community in natural forest differed from those in the hoop pine plantation (He, Xu & Hughes, 2005). These findings confirmed that revegetation with different tree species altered the soil fungal community diversity and composition. The significant contribution of different forest types for shaping the soil fungal community has been established by previous findings (Sun et al., 2016).

Relationship between fungal communities and soil environmental factors

Soil environmental factors demonstrated remarkable relationships with fungal diversity. The Simpson index and Shannon index were positively correlated with pH (Table 4). Similar results have been reported previously (Djukic et al., 2010; Liu et al., 2018b; Wang et al., 2015) that the diversity of the fungal community increased with soil pH value. In our study, soil Chao1 index, ACE index, and Shannon index significantly increased with the increasing C/N ratios (Table 4). In addition, ACE index and Chao1 index were significantly positively correlated with total P (Table 4), which was in accordance with a previous study reporting that fungal diversity is significantly affected by soil P-related factors (Liu et al., 2018a).

Just as the soil fungal diversity, soil environmental factors had greatly influenced on the fungal community composition. Previous studies have shown that soil physicochemical properties, such as soil moisture (Brockett, Prescott & Grayston, 2012), soil pH (Rousk et al., 2010), available soil nutrients (Lauber et al., 2008), soil total C (Yang et al., 2014), and C/N ratio (Lauber et al., 2008), strongly affected fungal communities. Moreover, our study also confirmed that the abundances of the most dominant fungal communities were significantly correlated with soil pH value. In addition, total C, total N, AN, and AP were also closely linked to the fungal community composition (Fig. 9; Table 5), which was consistent with other researches (Sun et al., 2016; Zhang et al., 2017b). Basidiomycota are generally sensitive to physic–chemical characteristic disturbance (Osono, 2007). In our study, the relative abundances of Basidiomycota was significantly negatively correlated with pH and AP, which was in contrast to the findings of previous studies (Tedersoo et al., 2014; Tian et al., 2017). In a previous study, soil with higher relative abundance of Ascomycetes has a higher pH value (Lauber et al., 2008). However, in our study, Ascomycota was not correlated with soil pH value. A relatively small pH range (4.89–5.70) was observed in our study, which might be difficult to ascertain such a correlation. Interestingly, the relative abundance of Ascomycota was positively correlated with total C, total N, and AN. In a recent study, Ascomycota abundance was associated with the content of soil organic matter (Sterkenburg et al., 2015). In our research, the abundance of Zygomycota was positively correlated with AP. This leads us to infer that soil AP was an important regulator of fungal communities, which is consistent with the findings of Dang et al. (2017). These results indicated that differential responses of soil fungal community composition to the different revegetation types largely dependent on soil physicochemical characteristics, highlighting the decisive role of soil physicochemical variables in altering fungal communities during vegetation restoration, which has also been stated previously (Kuramae et al., 2010).

Conclusions

Our results here showed that the different revegetation types would seem to play an important role in regulating soil characteristics in the same climate, especially between broadleaf forest and coniferous forest, which generated shifts in soil fungal community diversity and composition. Basidiomycota, Ascomycota, Zygomycota, and Rozellomycota were the predominant fungal community in Baishilazi Nature Reserve, and the relative abundances of these abundant fungal phyla varied significantly among the different revegetation types. The average Shannon index, ACE index, Chao1 index, and Simpson index were highest in J. mandshurica. The abundances of the most dominant fungal communities correlated significantly with soil pH, total C, total N, AN, and AP.

Supplemental Information

Supplemental Information 1 OTUs.

JM: Juglans mandshurica; QM: Quercus mongolica; CB: Conifer-broadleaf forest; LG: Larix gmelinii; PK: Pinus koraiensis. File 1 applied for data analyses and preparation for Figure 5, Figure 7, Figure 8, Table 3 and Table 4.

Click here for additional data file.

Supplemental Information 2 The relative abundance of fungal phylum.

JM: Juglans mandshurica; QM: Quercus mongolica; CB: Conifer-broadleaf forest; LG: Larix gmelinii; PK: Pinus koraiensis. File 2 applied for data analyses and preparation for Figure 4, Figure 9, and Table 5.

Click here for additional data file.

Supplemental Information 3 The relative abundance of fungal genus.

JM: Juglans mandshurica; QM: Quercus mongolica; CB: Conifer-broadleaf forest; LG: Larix gmelinii; PK: Pinus koraiensis. File 3 applied for data analyses and preparation for Figure S1, Figure 6, and Table 5.

Click here for additional data file.

Supplemental Information 4 Soil physical-chemical characteristics.

JM: Juglans mandshurica; QM: Quercus mongolica; CB: Conifer-broadleaf forest; LG: Larix gmelinii; PK: Pinus koraiensis. File 4 applied for data analyses and preparation for Figure 2, Figure 9, Table 2, Table 4 and Table 5.

Click here for additional data file.

Supplemental Information 5 The data of observed species corresponding to the observed OTU number.

JM:Juglans mandshurica; QM:Quercus mongolica; CB: Conifer-broadleaf forest; LG:Larix gmelinii; PK:Pinus koraiensis. File 5 applied for data analyses and preparation for Figure 3.

Click here for additional data file.

Supplemental Information 6 The distribution of partial sequences of fungal ITS gene at genus level.

JM: Juglans mandshurica; QM: Quercus mongolica; CB: Conifer-broadleaf forest; LG: Larix gmelinii; PK: Pinus koraiensis.

Click here for additional data file.

Additional Information and Declarations

Competing Interests

Author Contributions

Ethics

Data Availability

The authors declare that they have no competing interests.

Jiaojiao Deng performed the experiments, analyzed the data, contributed reagents/materials/analysis tools, prepared figures and/or tables.

You Yin analyzed the data, authored or reviewed drafts of the paper.

Jiyao Luo conceived and designed the experiments, performed the experiments.

Wenxu Zhu prepared figures and/or tables, authored or reviewed drafts of the paper, approved the final draft.

Yongbin Zhou conceived and designed the experiments, contributed reagents/materials/analysis tools.

The following information was supplied relating to ethical approvals (i.e., approving body and any reference numbers):

The Baishilazi Nature Reserve approved the field experiments (Ethical Application Ref: 20170628-7).

The following information was supplied regarding data availability:

Raw data is provided in the Supplemental Files.

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
