# Peer review of "Different revegetation types alter soil physical-chemical characteristics and fungal community in the Baishilazi Nature Reserve"

_PeerJ, doi:10.7717/peerj.6251_

## Round 0.1 · original submission · Major Revisions

The topic is well-received by 3 reviewers, however all are concerned with the writing, in terms of grammatical errors, lack of description of methods and an unclear discussion. The paper needs to be thoroughly revised so that it can be fully understood by international audiences.

·

Basic reporting

The paper have an important comparison of fungi community structure in different types of re-vegetation forests. Even with the limitation of replicates (three for each kind of re-vegetation forest) the author showed a big difference in fungi communities among the places. However their conclusion about re-vegetation types been the strong factor determine fungi composition was not explicit tested, other factor as bacterial composition or soil treatment was not discussed.

Experimental design

More explanation is needed about the soil collection, the authors said that they collect 15-20 sub-samples in each plot but not explain how these points were selected. Also they do not explain why the variation 15-20 sub-samples. Furthermore the analysis are badly explained, with confusion about software and packages, poor description of methods (e.g. which dissimilarity matrix was used in NMDS analysis) and no referenced. The author need re-write this part with more details, correct use of packages and software and cite the author that have a big work creating these tools.

Validity of the findings

The finds were interesting, however many test with many variables were conducted with very little replicates. I understand that all studies have sampling design limits but the author need at least discuss these limitations.

Additional comments

The study is nice, but the writing need to be improved, the analysis need to be re-write with detail and clarity and the discussion need to access the limitation of the study.

Reviewer 2 ·

Basic reporting

Basic reporting: In this work Deng and collaborators evaluated the effects of different revegetation types on soil physical-chemical characteristics and fungal community in the Baishilazi Nature Reserve using ITS rRNA Illunima Miseq high- throughput sequencing. I commend the authors for the excellent experimental design and extensive data collection. Unfortunately, the paper suffers greatly from poor language usage. Please pay special attention to grammar and syntax of the manuscript. There are several parts in the paper that need almost complete rewriting. I strongly suggest using an English language editing expert individual (English native speaking) or use a relevant commercial service.

In addition, the discussion needs to be improved as well. I suggest the authors to further examine former similar work and to interpret their results in terms of the impact of different revegetation types on the fungal community, as this section of the paper seems to be merely descriptive and lacks interpretation of their results (from ITS sequencing, soil physicochemical properties, and revegetation types). Also, several diversity indexes were calculated, and not appropriately considered. Why did you choose to estimate alpha diversity indices such as Chao1, ACE, Shannon, and Simpson? What distinctive traits of the community may be evaluated through each of these? Please explain in the methodology and review in the discussion.

Experimental design

Experimental design: The experimental design and the methods are acceptable for the most part. However, the appropriate software citation is lacking (Data Analysis, lines 145-148). Please include proper citation for QIIME, including the used version. The same for R, including the name (and proper citation) of the used packages.

Validity of the findings

Validity of the findings: I feel that the authors made appropriate conclusions for their work. However, it seems that ITS sequences were not deposited in a relevant database, and hence are not publicly available. According to the guidelines of PeerJ this is mandatory for publication. Besides, data availability is indispensable in order to ensure the reproducibility this investigation, and very helpful for future works. So, the authors are urged to provide accession numbers for their dataset, demonstrating its availability in a public relevant repository.

Additional comments

Comments for the author: This manuscript adds important and interesting information to the knowledge of soil fungal communities and their response to environmental fluctuations. The investigators in this paper did an exceptional amount of work that I feel should be mentioned. The science and conclusions appear sound and are interesting. However, before being considered for publication I feel that the manuscript needs to be importantly improved in English language usage, and carefully revised for careless errors (some examples include capitalization mistakes e.g. 10th line of the abstract, misspelling of Pearson statistical index, among several others). Also, as explained above, the discussion needs to be improved in order to enhance the contribution of this work. Lastly, it is crucial for the authors to make their datasets publically available.

Reviewer 3 ·

Basic reporting

Comments for the manuscript entitled “Different revegetation types alter soil physical-chemical characteristics and fungal community in the Baishilazi Nature Reserve”. The authors investigated the relationship between soil fungal community composition and soil properties to five re-vegetation areas with different dominant species (JM = Juglans mandshurica, QM = Quercus mongolica, CB = conifer-broadleaf forest, LG = Larix gmelinni, PK = Pinus koraiensis) at the Baishilazi Nature Reserve in the mountainous region in China.
It addresses one objective- to relate the soil fungi community composition with the type o re-vegetation- specifically, shifts in the abundance and composition of soil fungal communities to changes in soil properties.
The authors found interesting patterns for the relationship between soil fungi community composition with the type of re-vegetation, but some methodological aspects are unclear some. For example, What do they mean with available nitrogen, ammonium plus nitrate? and What technique did they use to determine available phosphorus or instruments to determine the soil total carbon?. How did they determine diversity or calculate the overall index? The above makes the interpretation and scope of results difficult to understand.
In general, the manuscript shows interesting data and results. It is relatively well written, although the verb conjugations and several details could be examined. The authors can improve the material and methods section. Also, it is very important to highlight that the plant-soil-microorganisms relationships are feedback relationships and it is difficult to determine the cause-consequence.
The introduction shows a good theoretical context, and the literature is relevant. The methods were not described with sufficient detail.

Experimental design

The research objective is well defined and it is relevant. The experimental design seems adequate, however the analysis of the relationship between soil properties and diversity of the fungi community could be improved for better results.

Validity of the findings

The analyses between diversity and soil properties can be improved if instead of the Pearson´s rank correlation coefficients (Table 4), an analysis of main components (PCA) was made with the set of soil properties and vegetation types. Also, regression analysis could be done with the resulting scores of the PCA. Where, the response variable would be each diversity index and the independent variable would be the analysis score of PCA.

Additional comments

L35-37 In sentence “Our results suggested that coniferous forest (LG, PK) and conifer-broad forest (CB) had reduced soil total C, total N, and Available N compared with broad-leaved forest (JM, QM)”, “Available N” should be “available N”.

L37-38 List of index could be as “Shannon, ACE, Chao 1 and Simpson index” instead of “Shannon index, ACE index, Chao 1 index and Simpson index”.

L44-46 In the sentence “The results of canonical correspondence analysis (CCA) shown that the soil environmental factors, such as soil pH, total C, total N, available N and available P had greatly influenced on the fungal community structure.”, “shown” should be “showed”.

L78-80 “work” should be “studies”
L81 “Different” should be “different”
L82 “Directly”, could be “Specifically”
L83 “nutrient” should be “nutrients”
L84-86 “was” should be “were”
L106-107 Are the symbols in the geographic coordinates correct?
L108 I do not understand the “hm2” units
In “Site description”, information about soil type according to the FAO classification is missing.
In “Material and Methods” section, information about determination of soil properties variables and diversity index calculation is missing.

L160 It is important to mention that the authors refer to nitrogen and available phosphorus, so it is necessary to include the analysis description of soil properties in the materials and methods section.

It is advisable to add a space between the values and the units that are reported in the results section and in all manuscript.

The discussion section can be improved if values of eco-physiological characteristics of plant species are included to understand the possible routes of their relationship with soil properties.

L262-263. Alkaline soil? Soils are very acidic, alkaline would be close to a pH of 8
L263-265. It would be advisable to add references
L270-272 It would be advisable to add the biological or ecological meanings of these relationships with the diversity index.
L279. It is recommended to remove the "and" for the best understanding of the idea.
L281 What do the authors mean with “their history of evolution”?

The analyses between diversity and soil properties can be improved if instead of the Pearson´s rank correlation coefficients (Table 4), an analysis of main components (PCA) was made with the set of soil properties and vegetation types. Also, with the resulting scores of the PCA, regression analysis could be done, where, the response variable would be each diversity index and the independent variable would be the analysis score of PCA.

Table 1 can be improved if the authors change “Types” in column #1 to something like “Vegetation types” or “labels of main species in re-vegetation areas” , and “Dominant Vegetation” to “Dominant species”, in “(m)” they could include “(m above sea level)”, and “Forest Type” to “Forest type”. The note at the end of the table can be place in the table description.

Table 2 can be improved if the authors use the same units for the total elements (g / kg) and for the available ones (mg / kg). The note at the end of the table can be place in the table description or (see table 1 for abbreviations).

Figure 5 does not seem very clear to me, but I am not an expert in the interpretation of this tool. I do not understand what the authors refer to "the relative abundance of the top 50 generates identified in the bacterial communities of the soils".

---

## Round 0.2 · Minor Revisions

The paper has been reviewed by 2 of the original reviewers. While the paper has much improved, there are still some snags and clarification that can be fixed. Clarification of the methods and analysis of data would make this paper better read.

·

Basic reporting

The authors improved the language and explain better methods and results, however there are yet problems in these sections. There are problems with reference style, the wrong citations, results showed without methods be presented (e.g. Principal Component Analysis line 201). The authors use a lot the word "significant" without show any test. Also the replicate are very limited, 15 plots in the total, and there is a high number of test. For instace, in the table 4, if Bonferrone correction was applied the significance level might be of p < 0.0017 (alpha/number of tests). The fifures 12 - 16 are repetitive of figure 1, and not cited in the text.

Experimental design

The experimental design is limited (three replicates for each revegetation type, 15 plots in the total), however it is hard have a good replicate number. I recomend the author, due the limitation of replicates, avoid multiple test, such as 4 different diversity index. Methods are yet with problems. There is confusing in the place where the analysis should be provided. There is no description of PCA in the methods, the dissimilarity index used in the NMDS just appear in the results and there is wrong way to cite reference (e.g. R software, that should be in this way - R v3.3.2 (R Development Core Team, 2017)). Title of figure 2 is wrong, there is Principle and the correct is the Principal.

Validity of the findings

The main problem of the valit of the findings is the replicate number. The results are sometimes dense and the discussion is a little confuse due the language. For exemple: "The composition of fungal community in terms of phyla and genera did not significantly differ among the different revegetation pathways, although the relative abundance values varied, probably as a result of different root residues and secretions produced by different tree species (Degrune et al., 2015). Although our research is limited due to the low number of replicates (three for each kind of revegetation forest), significant differences in fungal communities among different revegetation types were observed[But no difference in fungal composition, it is confuse, please explain better.]. " I couldn't understand what was different and what was not, more explanation is needed. Also the author use a lot the word "significant" when no test was performed or presented.

Reviewer 2 ·

Basic reporting

The authors have done a good job revising the manuscript, and have addressed all my previous suggestions. For this revised version, I have a few comments in relation to figures and tables and their legends:
• Figure 1: Letters A and B in red are very hard to read. Please edit the image in order to improve this problem (or perhaps change the color, the font, or even add a shade).
• Figure 4: Revise the use of italics. The label “other” should not be in italics.
• Figs. 7 and 9, where genus names should be written using italics.
• Please revise careless typos in figures labels. For example, missing spaces before “%” of the y-axis in Figs 4 and 5; and before the parenthesis in Fig. 10 (both axis).
• Figure 9: It is not mentioned what does panel “A” and “B” represent. Also, some labels (letters), corresponding to fungal taxa are jumbled (and therefore, difficult to read).
• Tables 2 and 3: It is not mentioned what does the letters (a, b, and c) after the values represent.
• Table 5: Legend “Person’s rank correlations between the relative abundances of dominant bacteria groups and available edaphic factors” does not correspond to the table, as this table presents fungal data as well.
• Panels in Figure 1 are not mentioned in the manuscript (within the text).
• Are files: peerj-30722, peerj-30722-File_2, peerj-30722-File_3, peerj-30722-File_4, peerj-30722-File_5 supplementary material? These are not mentioned in the manuscript. If they are intended as supplementary material pleas include a legend for each.
• It is desirable that the authors complete the information presented in Tables (peerj-30722-File_2, peerj-30722-File_3), with the following data (one line per sequence): Culture name, site
3, date of sampling, the result of best blast hit: species name with % of identity for the best coverage, species name retained in the present study.

Experimental design

No comments.

Validity of the findings

No comments.

Additional comments

The manuscript has greatly improved, I acknowledge the authors for their great effort. In principle, this study is relevant and expands knowledge. However, I still consider that it could be presented better. Particularly, redundant figures could be removed (e.g. Fig. 1 and Figs 11-16 are showing the same thing; I also feel that not all the barplots on the diversity results at a genus and phyla level are necessary as they don't add significantly to the information, so some of them could be moved to supplementary?).

---

## Round 0.3 · Minor Revisions

The paper has been revised but there are still few snags regarding statistical analysis. Please check throughout the manuscript.

The authors state that they did PCAs, but they don't give the software package that they used.

In the paper, often significance of P = 0.000 was quoted, which is impossible (something like P < 0.001 would be accurate and more acceptable as nothing is 100% certain/uncertain).

Abstract , L. 281, and L. 216 should be Pearson correlation analysis or Spearman rank correlation analysis?
I don't think there is a Pearson rank correlation analysis.
L. 284 etc "r" should be italicised

---

## Round 0.4 · accepted · Accept

The paper has been revised and is good for publication

#